# Upconversion Nanoparticle-Based Fluorescent Film for Distributed Temperature Monitoring of Mobile Phones’ Integrated Chips

**DOI:** 10.3390/nano13111704

**Published:** 2023-05-23

**Authors:** Hanyang Li, Miao Yu, Jichun Dai, Gaoqian Zhou, Jiapeng Sun

**Affiliations:** College of Physics and Optoelectronic Engineering, Harbin Engineering University, Harbin 150001, China

**Keywords:** upconversion nanomaterials, ratiometric thermometry, temperature sensing, polymer composite film, integrated chip temperature measurement

## Abstract

As one of the most critical parameters to evaluate the quality and performance of mobile phones, real-time temperature monitoring of mobile phones’ integrated chips is vital in the electronics industry. Although several different strategies for the measurement of chips’ surface temperature have been proposed in recent years, distributed temperature monitoring with high spatial resolution is still a hot issue with an urgent need to be solved. In this work, a fluorescent film material with photothermal properties containing thermosensitive upconversion nanoparticles (UCNPs) and polydimethylsiloxane (PDMS) is fabricated for the monitoring of the chips’ surface temperature. The presented fluorescent films have thicknesses ranging from 23 to 90 μm and are both flexible and elastic. Using the fluorescence intensity ratio (FIR) technique, the temperature-sensing properties of these fluorescent films are investigated. The maximum sensitivity of the fluorescent film was measured to be 1.43% K^−1^ at 299 K. By testing the temperature at different positions of the optical film, distributed temperature monitoring with a high spatial resolution down to 10 μm on the chip surface was successfully achieved. It is worth mentioning that the film maintained stable performance even under pull stretching up to 100%. The correctness of the method is verified by taking infrared images of the chip surface with an infrared camera. These results demonstrate that the as-prepared optical film is a promising anti-deformation material for monitoring temperature with high spatial resolution on-chip surfaces.

## 1. Introduction

In recent years, the continuous development of science and technology has made mobile phones a necessity for personal and global wireless communication [1,2]. With the advent of large-scale integrated circuits and the advancement of consumer electronics, there is a current trend in industry to integrate more and more functions into mobile phones while continuing to reduce weight and rapidly minimize device feature sizes [3,4]. Higher integration and smaller size lead to a significant increase in power density, which makes it important to achieve high-precision, high-sensitivity, and reliable temperature measurement of integrated chips at various scales [5]. The main reason is that elevated temperatures may increase circuit delays, while chips are susceptible to excessive temperature and shorten chip life [6,7,8]. The traditional thermometers that obtain the thermal behavior of chips, such as liquid glass thermometers and various types of electronic thermometers, cannot meet the requirements of harsh environments (high electromagnetic field interference, corrosive environments, etc.) and small spaces, real-time functioning, and rapid temperature measurement, resulting in inaccurate monitoring signals [9,10,11]. When measuring the temperature of chips, traditional infrared measurement methods are known to suffer from several limitations, including destructiveness, low spatial resolution, and limited accuracy [12,13]. Therefore, the real-time monitoring of chip temperature during normal operation and obtaining temperature distribution during safe operation are crucial measures to ensure the reliability of the chip.

Ratiometric optical thermometers have gained extensive attention in the field of high-precision and high-sensitivity temperature measurement due to their ability to accurately measure temperatures under conditions of power fluctuation and luminous loss [14,15,16,17,18,19,20]. Among the ratiometric optical thermometers, those based on Er^3+^/Yb^3+^ co-doping in different host matrices are the most widely studied, as Er^3+^ has abundant ladder-like arranged energy levels and Yb^3+^ has a sensitization effect that can produce strong green-upconversion luminescence (UCL) [20,21]. By exciting a 980 nm laser, the transition of the ^2^H_11/2_ → ^4^I_15/2_ and ^4^S_3/2_ → ^4^I_15/2_ energy levels can occur, thereby producing an upconversion luminescence fluorescence intensity ratio (FIR) [22,23,24]. FIR technology establishes the relationship between temperature and optical signals by taking advantage of the excellent thermal coupling characteristics between the ^2^H_11/2_ and ^4^S_3/2_ levels of Er^3+^, thereby allowing it to achieve high-quality temperature sensing under various extreme conditions, such as high voltage and strong electromagnetic fields, to reduce the influence of external environmental interference in the process of fluorescence recording [25,26,27]. Core–shell NaYF_4_:Er^3+^/Yb^3+^@NaYF_4_ upconversion nanoparticles are highly regarded for their exceptional fluorescence efficiency and strong temperature dependence, positioning them as promising materials for the development of high-precision and high-sensitivity temperature sensors [28,29].

Combining upconversion materials that can convert thermal radiation into light signals with polymer materials, which are materials with a polymer structure, results in a composite material with dual functionality that can be used for high-precision temperature measurements. This composite material has advantages such as high spatial resolution, high sensitivity, and non-contact measurement [30,31,32,33]. In the realm of microelectronics, high spatial resolution temperature distribution is vital in the real-time monitoring of circuit temperature and offering feedback, which can improve circuit reliability and extend the lifespan of electronic components [34]. Consequently, the development and implementation of UCNP/polymer optical films as distributed temperature sensors present a promising avenue for future research.

In this study, Polydimethylsiloxane (PDMS) was chosen as the polymer host due to its high light transmittance and simple curing process. β-NaYF_4_: Er^3+^/Yb^3+^@NaYF_4_ was synthesized via the high-temperature decomposition of lanthanide chloride precursors in the presence of high-boiling-point organic solvents and surfactants. The resulting product exhibits bright green upconversion fluorescence, which can be observed by the naked eye upon excitation with a 980 nm laser. Optical films with photothermal properties were fabricated, containing β-NaYF_4_: Er^3+^/Yb^3+^@NaYF_4_ UCNPs/PDMS. The maximum sensitivity of the optical film was measured as 1.43% K^−1^ at 299 K. This result indicates that the fabricated optical film can be used for temperature monitoring. It is worth mentioning that the film maintained stable performance even under pull stretch up to 100%. Experiments show that the film can be used to monitor the temperature of mobile phone chips in real-time and achieve a high-spatial-resolution measurement of distribution temperature. The reported polymer-based optical temperature measurement films have great application potential in the temperature measurement of integrated circuits and mobile phone chips.

## 2. Materials and Methods

### 2.1. Chemicals

All chemical reagents were obtained from Aladdin (Beijing, China) and used without further purification: ammonium fluoride (NH_4_F, >98%), sodium hydroxide (NaOH, >98%), 1-octadecene (ODE, 90%), oleic acid (OA, 90%), yttrium chloride (YCl_3_, 99%), ytterbium chloride (YbCl_3_, 99%), erbium chloride (ErCl_3_, 99%), methanol, chloroform, cyclohexane, ethanol, and polydimethylsiloxane (PDMS SYLGARD 184).

### 2.2. Synthesis of NaYF_4_:Er^3+^/Yb^3+^@NaYF_4_ Nanoparticles

Synthesis of NaYF_4_:Er^3+^/Yb^3+^. We weighed and mixed 1 mmol of lanthanide chloride (ErCl_3_, YbCl_3_, and YCl_3_) with a ratio of 2:18:80, dissolved in 10 mL of methanol, added 15 mL of ODE and 10 mL of OA into a 100 mL three-neck flask, and then added the above methanol mixture. The reaction environment was purged with Ar gas flow to ensure that no oxygen was present during the entire reaction. The mixed solution was stirred continuously with a magnetic stirrer to ensure uniform heating. The three-neck flask was heated to 150 °C and maintained for 30 min. After cooling to room temperature, 13 mL of methanol solution containing 4 mmol of NH_4_F and 2.5 mmol of NaOH was added into the three-neck flask. The mixture was then heated to 150 °C and maintained for 30 min to completely evaporate the methanol in the mixture. The three-neck flask was then heated to 300 °C for 90 min. After heating, the reaction was stopped, and the sample was naturally cooled to room temperature. The upconversion nanoparticles were prepared by washing the mixture with anhydrous ethanol through centrifugation, sonication, and precipitation steps. The washing steps were repeated until the dispersed solution of the sample becomes clear and transparent. Finally, the obtained sample was dried at 60 °C for 24 h to obtain NaYF_4_:Er^3+^/Yb^3+^ nanoparticles.

Synthesis of NaYF_4_:Er^3+^/Yb^3+^@NaYF_4_. NaYF_4_:Er^3+^/Yb^3+^@NaYF_4_ was prepared using a similar strategy to the one described above. First, 15 mL of ODE, 10 mL of OA, and 1 mmol of YCl_3_ chloride were dissolved in 10 mL of methanol and placed into a 100 mL three-necked flask. Ar gas flow was continuously introduced to ensure an oxygen-free reaction environment. A magnetic stirrer was added and stirred continuously to ensure uniform heating of the mixed solution. The mixture was heated at 150 °C for 30 min. After cooling to room temperature, a 13 mL methanol solution containing 4 mmol of ammonium fluoride, 2.5 mmol of sodium hydroxide, and 1 mmol of NaYF_4_:Er^3+^/Yb^3+^ nanoparticles was added to the flask using a pipette. The flask was heated at 150 °C for 30 min to evaporate the methanol completely. The flask was then heated to 300 °C for 90 min and then cooled to room temperature. The resulting upconverted nanocrystals were washed with anhydrous ethanol using centrifugation, sonication, and precipitation steps. The washing process was repeated until the dispersed solution of the sample became clear. The obtained sample was dried at 60 °C for 24 h to yield NaYF_4_:Er^3+^/Yb^3+^@NaYF_4_ nanoparticles.

### 2.3. Fabrication of NaYF_4_:Er^3+^/Yb^3+^@NaYF_4_-PDMS Film

The specific preparation steps for the UCNPs/PDMS thin film were as follows. First, 0.5 g of PDMS main solvent was added to 5 mL of chloroform solution. Then, 0.005 g of UCNPs were dissolved in 5 mL of cyclohexane. The two above-mentioned components were mixed with ultrasonic treatment for 1 h and left to stand at 60 °C for 24 h until the cyclohexane and chloroform were completely volatilized to obtain a composite material. To the prepared composite, 0.05 g of curing agent was added. The composite was placed on a glass slide before the composite was cured, and the composite was covered with another slide. Due to adsorption and gravity, the composite was flat between the two glass slides. Then, the composite was cured in an atmosphere of 80 °C for 40 min. After curing, the composite was demoulded and cleaned to obtain a composite film. The presented fluorescent films have thicknesses ranging from 23 to 90 μm and are both flexible and elastic. For the specific process, see Appendix A.

### 2.4. Characterization

The size distribution and morphology of the as-prepared NaYF_4_: Er^3+^/Yb^3+^@NaYF_4_ UCNPs were measured with transmission electron microscopy (TEM). The crystallographic orientations of NaYF_4_: Er^3+^/Yb^3+^@NaYF_4_ UCNPs were analyzed and verified using high-resolution transmission electron microscopy (HRTEM) and X-ray diffraction (XRD, Cu Kα radiation). The morphology and distribution of UCNPs in polymer films (UCNP/PDMS) were investigated using scanning electron microscopy (SEM), TEM, and elemental mapping techniques. The microscopic images of the polymer films (UCNP/PDMS) were observed using an optical microscope equipped with a CCD camera. The upconversion Luminescence (UCL) spectra of UCNPs under 980 nm laser excitation were recorded using a spectrometer.

## 3. Results and Discussion

NaYF_4_: Er^3+^/Yb^3+^@NaYF_4_ is synthesized via the high-temperature decomposition of lanthanide chloride precursors in the presence of high-boiling-point organic solvents and surfactants. For this study, appropriate modifications were made to the original technique to improve the particle size distribution and crystal quality of the resulting nanoparticles. These modifications included changes to the concentration of the lanthanide chloride precursors and the reaction temperature and time. Under the excitation of a 980 nm laser, Yb^3+^ was excited to ^2^F_5/2_ after gaining energy as a sensitizer, followed by energy transfer to activate Er^3+^. After this, Er^3+^ emitted three emission bands corresponding to the ^2^H_11/2_ → ^4^I_15/2_, ^4^S_3/2_ → ^4^I_15/2_, and ^4^F_9/2_ → ^4^I_15/2_ [35,36]. The whole process can be seen in Figure 1a. Figure 1a shows a schematic diagram of the principle by which core–shell UCNPs can enhance upconversion luminescence. The TEM image of the synthesized NaYF4:Er3+/Yb3+@NaYF4 UCNPs and the size distribution of NaYF_4_: Er^3+^/Yb^3+^@NaYF_4_ are shown in Appendix A. According to the pictures, it can be clearly observed that the NaYF_4_: Er^3+^/Yb^3+^@NaYF_4_ UCNPs form a hexagonal structure. The size of the UCNPs measured 100 times shows that the average length of NaYF_4_: Er^3+^/Yb^3+^@NaYF_4_ is about 57.11 nm, and the width is about 29.56 nm. Clear lattice fringes and an overall rod-like structure can be seen in the inset of Figure 1b. The d-spacing of the two lattice fringes with the shortest distance is about 0.29 nm, which is located in the (101) plane of the β-NaYF_4_ crystal. In Figure 1c, the purple curve represents the diffraction pattern of the as-prepared NaYF_4_: Er^3+^/Yb^3+^@NaYF_4_, and the black straight segment parallel to the *y*-axis is the standard β-NaYF_4_ (JCPDS No. 16-0334). Through the analysis and interpretation of the XRD patterns and HRTEM images of the materials, UCNPs are shown to be nanomaterials with single crystal properties and a hexagonal structure. NaYF_4_: Er^3+^/Yb^3+^ and NaYF_4_: Er^3+^/Yb^3+^@NaYF_4_ exhibited green upconversion luminescence macroscopically under 980 nm laser excitation, and their corresponding upconversion fluorescence emission spectra are shown in Figure 1d. The upconversion fluorescence photos of NaYF_4_: Er^3+^/Yb^3+^ (left) and NaYF_4_: Er^3+^/Yb^3+^@NaYF_4_ (right) cyclohexane solutions under the same power of 980 nm laser excitation can be observed in the inset of Figure 1d, where it can be seen that the right side is brighter than the left side. The energy level transitions of UCNPs at ^2^H_11/2_ → ^4^I_15/2_ and ^4^S_3/2_ → ^4^I_15/2_ correspond to green emission at 525 nm and 540 nm, respectively. Core–shell UCNPs show 8.2 and 9.3 times higher intensity compared to core-only UCNPs at 525 nm and 540 nm energy level transitions, as calculated by fluorescence spectroscopy. The stronger upconversion fluorescence intensity of core–shell-structured UCNPs is attributed to the addition of an inert NaYF_4_ shell layer at the interface between the core and the shell. The NaYF_4_ shell layer can reduce surface defects and non-radiative energy transfer, thereby reducing surface-quenching effects. Additionally, the shell layer can serve as an additional protective layer to prevent oxidation and degradation of rare earth ions in the core. Moreover, the shell layer can also modulate the surface properties of the core–shell UCNPs, for example, altering the surface charge and increasing biocompatibility, thus presenting potential applications in biomedical imaging and therapy [37,38].

PDMS can be used as an optical waveguide material due to its excellent optical transparency and low refractive index (1.406). Its relatively low dielectric constant can reduce the loss of optical signals, making it suitable as a substrate or channel for optical devices [39,40]. Based on the above advantages, we used PDMS hybrid UCNPs to fabricate fluorescent film. The curing of PDMS is based on crosslinking reactions between the base and the curing agent. Through multiple experiments, it was found that when we chose 1 wt%, the UCNPs in PDMS can maintain good optical properties. The structural features of this optical film can be analyzed and determined by SEM, TEM, and elemental mapping. Figure 2a is a film (UCNP/PDMS) photograph taken by a digital camera. As shown, the film exhibits high definition in sunlight (top) and a distinct green emission (bottom) when excited by a 980 nm laser under darkfield conditions, demonstrating the excellent UCL performance of these materials. Figure 2b,c show the SEM images of the films (UCNP/PDMS) at different scales, from which it can be seen that the films form a planar structure with a smooth surface and uniform thickness (thickness ≈ 23 μm).

The TEM images of the polymer films at different resolutions are shown in Figure 2d,e. The image shows that the nanoparticles are relatively uniformly dispersed in the PDMS, but there are some clusters of varying sizes, which may be due to the van der Waals forces [41] and the electrostatic attraction [42] between the nanoparticles and their own molecules or atoms. As shown in Figure 2f–i, F, Y, and Yb elements were successfully distributed in PDMS. The distribution of complete elements is shown in Appendix A. The distribution of various elements showed that NaYF_4_:Er^3+^/Yb^3+^@NaYF_4_ was incorporated into PDMS.

The image shows that the nanoparticles are relatively uniformly dispersed in the PDMS, but there are some clusters of varying sizes, which may be due to the van der Waals forces [41] and the electrostatic attraction [42] between the nanoparticles and their own molecules or atoms.

The photothermal properties of NaYF_4_:Er^3+^/Yb^3+^@NaYF_4_/PDMS films were investigated to assess the optical temperature sensing capability of the prepared polymer films. A fiber was connected to a 980 nm laser (LWIRL980-7W, Laserwave, Beijing, China) as the pump light source, and the fiber tip was placed in close proximity to the polymer film to observe green emission in the local area at a power of 5 mW. UCL spectra were recorded using a portable spectrometer (QEpro, Ocean Optics, Shanghai, China) capable of measuring upconversion fluorescence emission. The temperature around the polymer film was controlled using a temperature control device with a resolution of 0.1 K that could read temperature data at any time. When the temperature was stable and the readings of the temperature control device were fixed, the spectrum was recorded. The specific equipment used to study the photothermal properties of polymer films is shown in Appendix A. Figure 3a shows the evolution of the upconversion fluorescence spectrum of the NaYF_4_:Er^3+^/Yb^3+^@NaYF_4_/PDMS film when the local temperature was increased from 299 K to 359 K. The images indicate that the upconversion luminescence peaks of NaYF_4_:Er^3+^/Yb^3+^@NaYF_4_ UCNPs are at 525 nm and 540 nm. As the temperature increases, the upconversion luminescence intensity at 540 nm gradually decreases, while the upconversion luminescence centered at 525 nm increases slowly. The specific intensity changes are shown in Figure 3b, and the rate of upconversion intensity changes in ^2^H_11/2_ → ^4^I_15/2_ is significantly smaller than that in ^4^S_3/2_ → ^4^I_15/2_. Due to the small energy difference existing between the ^2^H_11/2_ energy level and the ^4^S_3/2_ energy level, the ratio of the ^2^H_11/2_ → ^4^I_15/2_ transition intensity to the ^4^S_3/2_ → ^4^I_15/2_ transition intensity varies with temperature as the temperature increases [43,44]. This is the fluorescence intensity ratio (FIR) technique that can be used for temperature measurement. The population of the two thermal coupling energy levels ^2^H_11/2_ and ^4^S_3/2_ following the Boltzmann distribution law can be expressed as [33,43]
(1)$$FIR=\frac{I_{H}}{I_{S}}=Cexp\left(\frac{-\Delta E}{kT}\right)$$
in which *I*_H_ and *I*_S_ are the integrated intensities of the ^2^H_11/2_ → ^4^I_15/2_ and ^4^S_3/2_ → ^4^I_15/2_ transitions, respectively. *C* is a constant, Δ*E* represents the energy gap between ^2^H_11/2_ and ^4^S_3/2_ levels, *k* is the Boltzmann constant, and *T* is the temperature in Kelvin. Figure 3b shows the upconversion fluorescence intensity integral of the material, and notes that the *I*_H_ value fluctuates at 339 K, which may be caused by the instability of the excitation power. In the process of gradually increasing the temperature from 299 K to 359 K, the value of FIR gradually increased from 0.28 to 0.44. Equation (1) can be used to fit the relationship between temperature and FIR value more accurately. According to Figure 3c, it can be concluded that a regression coefficient (R^2^) fits all the measured points more accurately, and the fitting results show that the values of C and ΔE in this experiment are 19.47 and 916 cm^−1^, respectively. The value of ΔE shows that this is consistent with the Er^3+^ energy level difference.

The rate of change in FIR with temperature can be used to quantitatively characterize the temperature sensing performance of optical temperature-measurement materials in practical applications. The absolute change value of FIR when the temperature changes by 1 K and the relative change rate with respect to itself are usually defined as the absolute sensitivity (*S*_a_) and relative sensitivity (*S*_r_), respectively. For absolute sensitivity (*S*_a_), it can be understood as the rate of change in FIR with temperature, which is calculated as follows [45]:(2)$$S_{a}=\frac{dFIR}{dT}=FIR\left(\frac{\Delta E}{kT^{2}}\right)$$

Relative sensitivity (*S*_r_) is the normalization of absolute sensitivity (*S*_a_) relative to FIR, which is widely used in temperature measurement [43,46].
(3)$$S_{r}=\frac{1}{FIR}\frac{dFIR}{dT}=\frac{\Delta E}{kT^{2}}$$

The fitting curves of the values of *S*_a_ and *S*_r_ in the temperature range of 299–359 K are shown in Figure 3d. It is evident from the picture that the value of *S*_a_ is proportional to the temperature, with a maximum value of 0.57% K^−1^ at 359 K, while the value of *S*_r_ is inversely proportional to the temperature, with a maximum value of 1.43% K^−1^ at 299 K. The FIR values at different positions of the same membrane were measured, and the fitting curves of FIR and temperature at different positions were obtained, as shown in Appendix A. Experiments show that the photothermal properties of different positions of the same film are exactly the same, which makes it possible to use this polymer film to achieve the idea of a distributed temperature-measurement method. Table 1 compares the sensitivity of several Er^3+^ in different host materials. Compared with some previous reports, the film used for this measurement has higher sensitivity, indicating that it has better photothermal properties. The membrane was placed in environments of 30 °C, 30–40 °C, 30–60 °C, and 30–80 °C to record the FIR values cyclically, as shown in Figure 3e. This result indicates that the film has high reliability and repeatability.

PDMS is a flexible organic silicone material with good elasticity and deformability. In practical applications, PDMS may be subjected to external forces, such as deformation and stretching. Therefore, it is necessary to study its performance and response characteristics under deformation to ensure its reliability and accuracy in complex environments. This can provide practical performance data and references for relevant applications and can improve the design and performance of relevant instruments [31,51,52]. A polymer film of appropriate size was selected and fixed at both ends to apply opposing forces, as shown in Figure 4a. The same experimental method was used to record the photothermal characteristics of the film under different degrees of stretch.

Figure 4b shows the UCL spectra of the film under various degrees of stretching. It can be observed from the fluorescent spectrum that the UCL intensity of the film gradually decreases as the stretching strength increases, which may be due to the length of the light-attenuation path. However, according to Figure 4c, the effects of different stretching strengths at the same temperature have a negligible impact on the FIR value. This suggests that the deformation has a minimal impact on the experimental results when using a polymer film for temperature measurement. Figure 4d shows the relationship between the FIR values and temperatures under different degrees of deformation. The fitting curve of the FIR value and temperature in the stretched state also satisfies Equation (1) when compared with the case where there is no stretching. Furthermore, three other films of varying thicknesses were studied, and the FIRs obtained from these sensors in the 299–349 K range are shown in Appendix A. Through this series of experimental results, it is demonstrated that the photothermal properties of the NaYF_4_:Er^3+^/Yb^3+^@NaYF_4_/PDMS-based polymer film can be used as a deformation-resistant optical temperature sensor due to its high sensitivity and stability.

The NaYF_4_: Er^3+^/Yb^3+^@ NaYF_4_/PDMS film prepared, benefiting from the strong anti-electromagnetic interference ability of PDMS, can be used for temperature measurement in electronic devices. Here, the Exynos7420 chip in Galaxy S6 edge+ is selected as the main research object. The optical temperature measurement of an external pump source mainly measures the average temperature of the excitation zone, and high-spatial-resolution distributed temperature measurement can be achieved as long as the excitation zone is as small as possible. Using a fiber tip to guide the excitation light to the film to excite the fluorescence material can achieve high-sensitivity optical measurement on a small surface. The focusing principle of the fiber tip is similar to that of a lens, which can focus the light on a small area with a diameter of a few hundred nanometers without damaging the sample during measurement [53,54,55]. Therefore, a fiber optic tip with a top diameter of ≈2 μm was chosen to excite the upconversion fluorescence of the film under the condition of an additional laser at 980 nm. The most important part of distributed temperature measurement is the position of the fiber optic tip, which is firmly fixed on a 3D adjustment frame with a constant excitation angle. The device for measuring the temperature of the chip surface is shown in Appendix A. A UCL spectrum was recorded by moving the 3D adjustment frame every 10 μm, thereby achieving high-spatial-resolution temperature measurement, as shown in Appendix A.

Figure 5a shows a schematic of the surface temperature measurement on the chip. Figure 5b displays a micrograph of the fluorescent film on the chip surface excited by a 980 nm laser fibertip. Figure 5c shows the chip surface images before (left) and after (right) coating, with points A, B, and C selected for real-time monitoring of chip surface temperature. Figure 5d presents the temperature–time relationship of points A, B, and C during repeated on-off cycles. The phone temperature gradually rises within 30 s after the phone is turned on (pink region). After 50 s, when the phone was idle, the chip temperature stabilized (blank region). After a stable period, the phone was turned off at 200 s (blue region), and the temperature was measured again during the following 1700 s. The real-time emission spectra of the fluorescent film during the on-off cycles within the first 200 s are shown in Appendix A. The entire chip was divided into 7 × 7 regions, and four temperature measurements were taken at the center of each region to obtain the average temperature of the entire area. Figure 5e displays the chip surface temperature distribution measured by the fluorescent film. Figure 5f shows the chip surface temperature distribution measured by an infrared camera (Hikvision DS-TPH10-3AUF). By comparing the two sets of data, seven identical temperature collection points were extracted along the temperature-extraction line, as shown in Figure 5g. The correctness of the temperature measurement method was validated by comparing the experimental data with the infrared imaging results. It is evident that the temperature trend measured by the fluorescent-doped polymer film is entirely consistent with the one obtained from the infrared thermal imaging, which proves that the prepared film has excellent temperature-measurement performance. The distributed temperature monitoring of the chip surface with high spatial resolution as low as 10 μm has been successfully realized, providing more accurate and precise information for research and applications in various fields. This method has significant application value for the design, manufacturing, and performance improvement of microelectronic devices.

Comparing the temperature distribution of different types of mobile phone chips is of great significance for understanding their thermal performance and improving their design. Using the same measurement method as described above, the surface temperature distribution of the MTK6752 chip in the Vivo X5s, the Kirin970 chip in the Huawei p20, and the A10 chip in the iPhone 7plus under stable operation was measured, as shown in Appendix A. Table 2 compares the different surface temperature distributions of different types of mobile phone chips based on upconversion materials, providing valuable insights into the thermal characteristics of these chips. The Kirin970 chip’s temperature difference is at least 2.3 °C. In addition, the highest temperature of the Samsung Exynos7420 is 43.9 °C, while the lowest temperature of the Apple A10 is 34.7 °C. These findings contribute to improving the design of mobile phone chips to enhance their thermal performance and reduce energy consumption. As mobile phone chips become more powerful, their thermal performance becomes increasingly important for maintaining their reliability and lifespan. Future development scenarios will involve the use of advanced cooling technologies such as heat pipes and liquid cooling to effectively dissipate the heat generated by mobile phone chips. Developing more efficient and reliable cooling technologies is crucial for the design and performance of future mobile phones.

## 4. Conclusions

To summarize, a soft and stretchable fluorescent thermometric film has been developed successfully that can be used for the real-time monitoring of electronic integrated chips. The film is composed of fluorescent films (UCNP/PDMS) with thicknesses ranging from 23 to 90 μm, flat surfaces, and a uniform thickness that was manufactured using a simple method. The morphology of NaYF_4_: Er^3+^/Yb^3+^@ NaYF_4_ nanoparticles in PDMS was found to be unaffected, as confirmed by characterization. The Er^3+^-based films showed high sensitivity within the temperature range of 299–359 K, with a sensitivity coefficient of 1.43% K^−1^ at 299 K, when using FIR technology. Remarkably, the photothermal properties of this film were found to be unaffected by deformation, such as stretching, which is of great importance for further research on portable and wearable sensors. In addition, the prepared optical film has a high spatial resolution (10 μm) and is suitable for surface and distributed temperature measurements of four different types of mobile phone chips. Furthermore, the optical film’s small size, high sensitivity, and resistance to strong electromagnetic interference make it well-suited for use in microelectronic applications. It is an anti-deformation material that can monitor and provide real-time feedback on circuits.

## Figures and Tables

**Figure 1 nanomaterials-13-01704-f001:**
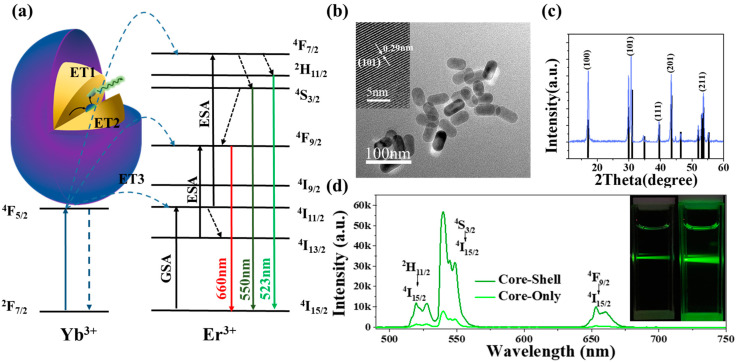
(**a**) Schematic diagram of the internal structure of the nanoparticle NaYF_4_:Er^3+^/Yb^3+^@NaYF_4_. Energy level diagram of NaYF_4_:Er^3+^/Yb^3+^@NaYF_4_ and possible up-conversion luminescence mechanisms. (**b**) TEM image of NaYF_4_:Er^3+^/Yb^3+^@NaYF_4_; the inset shows an HRTEM photograph of an individual NaYF4 microcrystal. (**c**) XRD pattern of the core–shell UCNPs (purple curve); the black line represents the standard diffraction pattern of β-NaYF_4_. (**d**) Upconversion fluorescence spectra of NaYF_4_:Er^3+^/Yb^3+^@NaYF_4_ (core–shell) and NaYF_4_:Er^3+^/Yb^3+^ (core-only). The inset shows UCL photographs in the mixed solution of NaYF_4_:Er^3+^/Yb^3+^@NaYF_4_ (core–shell) and NaYF_4_:Er^3+^/Yb^3+^ (core-only) excited by 980 nm laser, which were taken with a digital camera.

**Figure 2 nanomaterials-13-01704-f002:**
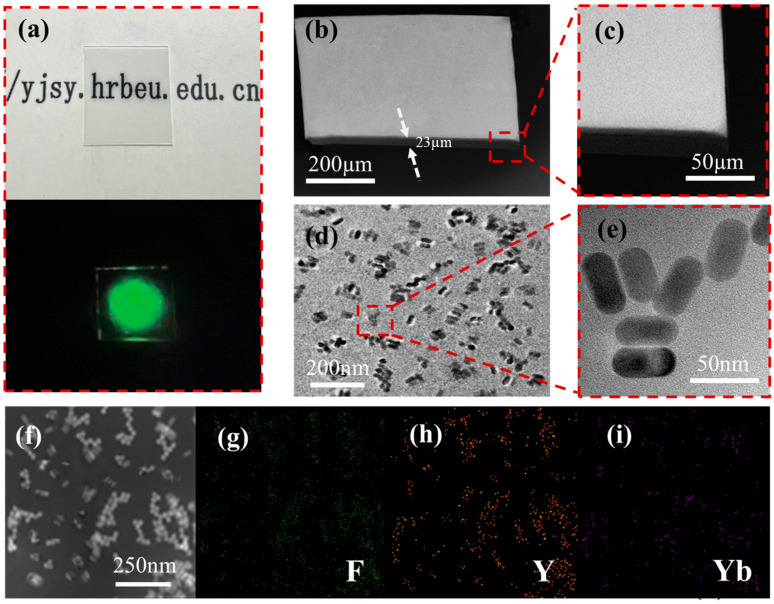
(**a**) Photographs of the as-prepared NaYF_4_:Er^3+^/Yb^3+^@NaYF_4_/PDMS, the upper image is a photo of the optical film under sunlight, and the lower image is a photo of the film excited by a 980 nm laser under darkfield conditions. (**b**,**c**) SEM images of thin films at different scales. (**d**,**e**) TEM micrographs of NaYF_4_:Er^3+^/Yb^3+^@NaYF_4_/PDMS at different scales. (**f**–**i**) Elemental mapping images of NaYF_4_:Er^3+^/Yb^3+^@NaYF_4_/PDMS compounds.

**Figure 3 nanomaterials-13-01704-f003:**
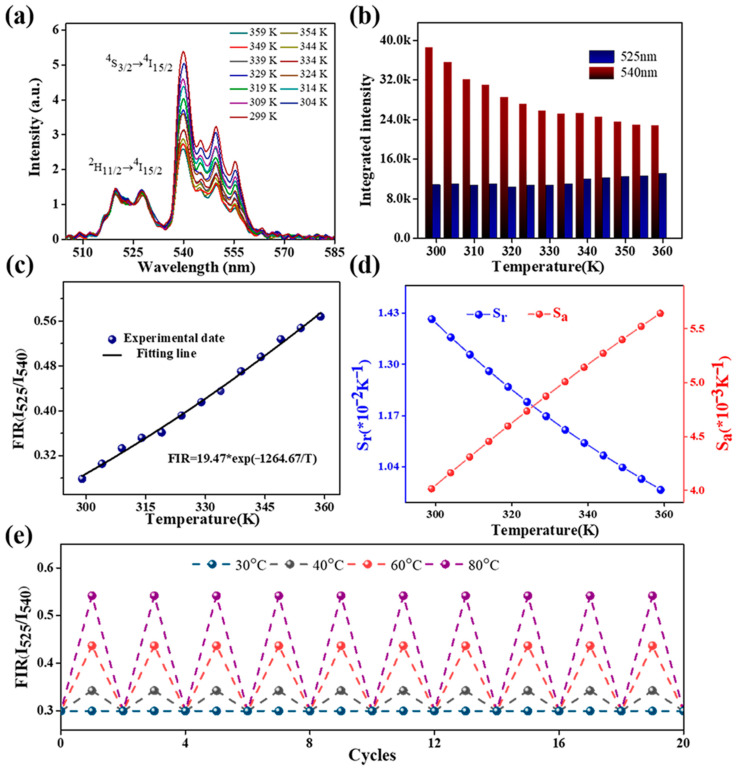
(**a**) The change in the upconversion fluorescence spectrum of the film under the excitation of 980 nm and the temperature range of 299−359 K. (**b**) Integral upconversion intensity versus temperature at 525 nm and 540 nm. (**c**) FIR versus temperature. (**d**) Temperature sensitivity versus temperature. (**e**) FIR values when the temperature is held constant at 30 °C and periodically varied between various settings at 30 °C (i.e., 40 °C, 60 °C, and 80 °C).

**Figure 4 nanomaterials-13-01704-f004:**
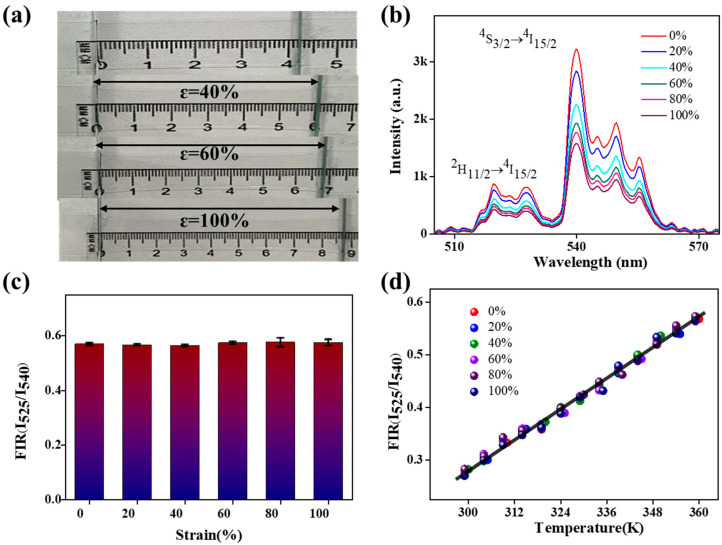
(**a**) The image is a photograph of the optical film under various tensile strains up to 100%. (**b**) UCL spectra of the film under stretching. (**c**) The influence of different stretching strengths on FIR(I_525_/I_545_) values at the same temperature. (**d**) Dependence of FIR(I_525_/I_545_) on the temperature (K), measured under various tensile strains up to 100%.

**Figure 5 nanomaterials-13-01704-f005:**
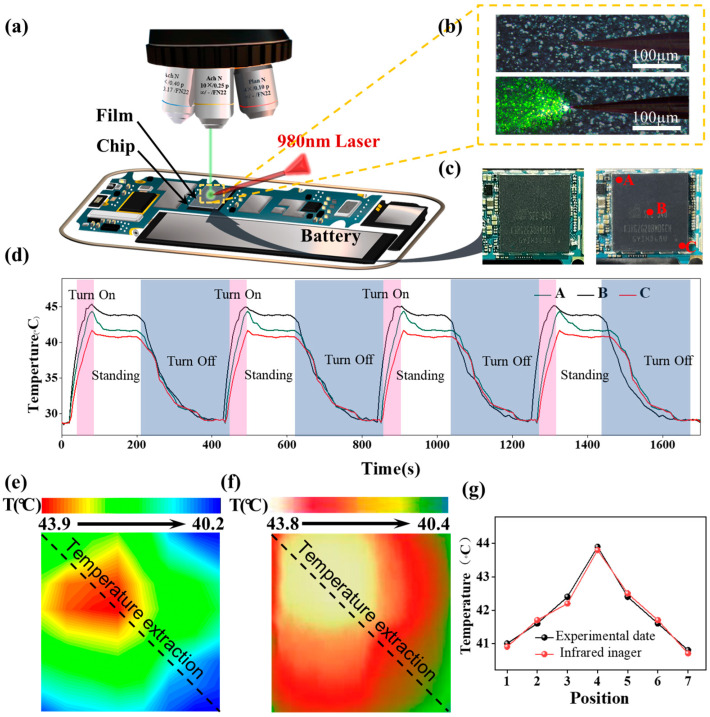
(**a**) Experimental equipment for temperature measurement. (**b**) Microscopic images of the fluorescence film on the surface of the chip excited by an external 980 nm laser fiber cone. (**c**) The actual image before and after the chip covers the film and three temperature measurement points A, B, and C are marked. (**d**) The temperature time relationship diagram of points A, B, and C in the chip during repeated on/off processes. (**e**) Fluorescent film measurement temperature distribution image. (**f**) Infrared image of the tested chip. (**g**) Comparison of experimental data and infrared image temperature curve.

**Table 1 nanomaterials-13-01704-t001:** Comparison of different optical-temperature-sensor-based upconversion materials.

Phosphor	Maximum Sensitivity(% K^−1^)	Temperature Range (K)	Ref.
NaBi(WO_4_)_2_:Yb^3+^/Er^3+^	1.24	298–373	[47]
BaGdF_5_:Yb^3+^/Er^3+^	1.28	298–681	[48]
Al_2_O_3_:Yb^3+^/Er^3+^	0.51	295–973	[49]
NaYF_4_:Yb^3+^/Er^3+^	1.68	258–423	[50]
NaYF_4_:Yb^3+^/Er^3+^@NaYF_4_	1.43	299–359	This work

**Table 2 nanomaterials-13-01704-t002:** Surface temperature distribution of different types of mobile phone chips.

Chip Type	Mobile Phone	Maximum Temperature (°C)	Minimum Temperature (°C)	Temperature Difference (°C)
A10	iPhone 7plus	34.7	32.0	2.7
Kirin970	Huawei p20	36.9	34.6	2.3
MTK6752	Vivo X5s	40.4	37.4	3.0
Exynos7420	Galaxy S6 edge+	43.9	43.9	3.7

## Data Availability

The data presented in this study are available on request from the corresponding author.

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
