# Peer review of "Upconversion Nanoparticle-Based Fluorescent Film for Distributed Temperature Monitoring of Mobile Phones’ Integrated Chips"

_nanomaterials, 2023, doi:10.3390/nano13111704_

Round 1

Reviewer 1 Report

This article suggests the use of PDMS films with upconversion thermosensitive fluorescent nanoparticles for monitoring the surface temperature of mobile phone integrated chips. 

The presentation and the work are excellent.This work is original and I recommend publications as it is.

Author Response

Manuscript ID: nanomaterials-2400309

TITLE: Upconversion nanoparticle-based fluorescent film for distributed temperature monitoring of mobile phone integrated chips

Dear Editor,

Thank you very much for your work on our manuscript. We also appreciate the editor and the reviewer’s carefulness, conscientiousness, and the broad knowledge on relevant research fields, since they have given us a number of beneficial suggestions.

Thank you for the positive recognition and acknowledgement of our work. We are honored to receive such a high evaluation from the reviewers. If you have any questions, please contact with us without any hesitation. We hope that our manuscript has meets the publication standards of Nanomaterials, as well as those of its special issue on Luminescent Nanomaterials: Functional Design, Advantages, and Applications.

Thank you very much for your time and effort on this paper and we look forward to hearing from you soon again.

Sincerely,

Hanyang Li

Reviewer 2 Report

The paper entitled “Upconversion nanoparticle-based fluorescent film for distributed temperature monitoring of mobile phone integrated chips” contains very useful results about temperature measurement of chip surfaces using a surface layer made from thermosensitive nanoparticles containing polydimethylsiloxane films.

The paper, however, is written without large attention, there are a lot of typographical mistakes and some non-usual /erroneous expressions, and some questionable parts in the text. Therefore, the paper has to be revised before accepting as a publication in “Nanomaterials”.

For example,

p.2. par.3., line 3 “..by thermal decomposition technology…”, here it should be given what will be thermally decomposed to get these materials?

p.2. The last par. Line 3, YCl3, subscript

p.3. The first par., line 4, and par. 2, line 4,  “Ar airflow”, argon may not be airflow, only gas flow (gas stream).

p.3. par. 1, line 8 , …s NaOH”, what is the “s” ?

p.3. par. 3., “…two drugs”, I think, It had better written “two components”

p.4. par. 1., ….”..with appropriate modifications using the thermal decomposition technique”. It should be given, what was the original technique (a bit more than “thermal decomposition technique”, e.g. starting materials, etc.,), and what kind of modifications have been done?

 p.4. par. 1., line 30, NaYbF4, subscript

p.6. If the particles are arranged due to the mentioned forces, some evidence should be given about that. I can see only that there is a random distribution in the matrix. Please explain the reasons and evidence for the controlled arrangement. The picture given shows an arrangement with very different orientations.

p.6. par. After eqn. 1., line 6, Kelvin is K and not “k”

The maximum sensitivity of Yb/Er-doped NaYF4 is better than the used composite,a nd the temperature range is also wider. It should be given some explanation/discussion why the author selected the @NaYF4 type composite and not the simple doped one, explaining the results of Table 1. ?

In Figure S2 only black squares can be seen, it is not possible to follow the distribution of elements.

The topic and results are interesting, and after revision, I think, it will fit to “Nanomaterials”.

Reviewer 3 Report

The MS nanomaterials-2400309 represents carefully performed and well written work, which due to its practical significance diserves publishing after minor revisions. It is required to specify the curing agent applied in the work. It is not clear why in the experimental part, describing the synthesis of composite films, the author calls the components of the film drugs?

The work represents very nice combination of nanotechnological and technological approaches, and it diserves acceptance after the abovementioned minor revisions.

Reviewer 4 Report

Authors elaborate and study a PDMS film containing thermosensitive upconversion nanoparticles and discuss its potential for temperature monitoring, in view of application to mobile phone and other electronic integrated chips. This comprehensive experimental work and very well-written text will certainly deserve publication in Nanomaterials after consideration of the few following points.

1. The references 1, 2, 3, 5 and 6 have no link with the argument they are supposed to support. These references concern various materials developed for local temperature monitoring and the text describes tendencies and technical issues of mobile phone integration. I suggest to keep the text and to replace the references.

1a. In case where the references 3 and 5 were re-used at another place of the article, I draw attention to errors in the reference list: the title of reference 3 is missing and the title of reference 5 is wrong (in fact, this title corresponds to "Puddu et al., Adv. Healthcare Mater. 2015, 4, 1332").

2. In paragraph 2.2, what is the meaning of "s" in "2.5 mmol of s NaOH"? Do not use this abbreviation, please.

3. In paragraph 2.3, indicate the range of film thicknesses, please

Round 2

Reviewer 2 Report

The requests have been fulfilled and the questions have been answered. The paper improved a lot, thus I can suggest accepting it. 

Author Response

Thank you for your attention to our manuscript entitled “Upconversion nanoparticle-based fluorescent film for distributed temperature monitoring of mobile phone integrated chips” (Manuscript ID: nanomaterials-2400309). We also appreciate the editor and the reviewer’s carefulness, conscientiousness, and the broad knowledge on relevant research fields, since they have given us a number of beneficial suggestions.

Thank you for the positive recognition and acknowledgement of our work. We are honored to receive such a high evaluation from the reviewers. If you have any questions, please contact with us without any hesitation. We hope that our manuscript has meets the publication standards of Nanomaterials, as well as those of its special issue on Luminescent Nanomaterials: Functional Design, Advantages, and Applications.

Thank you again for the time and effort you have put into this paper.

Sincerely,

Hanyang Li
